# The RODI mHealth app Insight: Machine-Learning-Driven Identification of Digital Indicators for Neurodegenerative Disorder Detection

**DOI:** 10.3390/healthcare11222985

**Published:** 2023-11-19

**Authors:** Panagiota Giannopoulou, Aristidis G. Vrahatis, Mary-Angela Papalaskari, Panagiotis Vlamos

**Affiliations:** 1Bioinformatics and Human Electrophysiology Laboratory, Department of Informatics, Ionian University, 49100 Corfu, Greece; c16gian@ionio.gr (P.G.); aris.vrahatis@gmail.com (A.G.V.); 2Department of Computing Science, Villanova University, Villanova, PA 19085, USA; map@villanova.edu

**Keywords:** neurodegenerative disorders, mHealth apps, machine learning, digital indicators

## Abstract

Neurocognitive Disorders (NCDs) pose a significant global health concern, and early detection is crucial for optimizing therapeutic outcomes. In parallel, mobile health apps (mHealth apps) have emerged as a promising avenue for assisting individuals with cognitive deficits. Under this perspective, we pioneered the development of the RODI mHealth app, a unique method for detecting aligned with the criteria for NCDs using a series of brief tasks. Utilizing the RODI app, we conducted a study from July to October 2022 involving 182 individuals with NCDs and healthy participants. The study aimed to assess performance differences between healthy older adults and NCD patients, identify significant performance disparities during the initial administration of the RODI app, and determine critical features for outcome prediction. Subsequently, the results underwent machine learning processes to unveil underlying patterns associated with NCDs. We prioritize the tasks within RODI based on their alignment with the criteria for NCDs, thus acting as key digital indicators for the disorder. We achieve this by employing an ensemble strategy that leverages the feature importance mechanism from three contemporary classification algorithms. Our analysis revealed that tasks related to visual working memory were the most significant in distinguishing between healthy individuals and those with an NCD. On the other hand, processes involving mental calculations, executive working memory, and recall were less influential in the detection process. Our study serves as a blueprint for future mHealth apps, offering a guide for enhancing the detection of digital indicators for disorders and related conditions.

## 1. Introduction

The human brain undergoes structural changes throughout life, impacting individuals’ cognitive abilities. Cognitive decline in older adults is particularly worrisome due to its medical and socioeconomic implications. Dementia, a prevalent condition among the elderly, affects approximately 5% of the elderly worldwide, while it is projected to increase due to population aging and longer life expectancy [1,2,3]. Biological and environmental factors play a significant role in determining outcomes, ranging from healthy cognitive aging to the development of Neurocognitive Disorders (NCDs), which can manifest years before clinical symptoms appear [4,5]. Moreover, cognitive performance does not follow a specific age pattern for peaking, as the onset and progression of cognitive decline can vary [6]. Interventions aimed at healthy adults with typical cognitive functioning are essential for promoting healthy aging and potentially reducing the risk of cognitive decline. By harnessing brain plasticity and cognitive reserve, these interventions enhance cognitive skills and resilience throughout the aging process [7,8].

The evaluation, diagnosis, and monitoring of cognitive function have traditionally been conducted in clinical settings using standardized neuropsychological assessments and interviews. However, barriers such as limited access to healthcare services and lengthy waiting lists can pose challenges for individuals, potentially deterring them from seeking necessary medical care. These deterrents restrict the vital early detection of cognitive deficits and the timely implementation of the appropriate cognitive interventions [9]. Moreover, regular monitoring is essential for accurately assessing changes in cognitive status, enabling the provision of suitable intervention strategies and support to individuals.

The integration of new sensors in mobile and wearable digital technology has revolutionized the development of a diverse range of assistive applications, addressing various needs and challenges across different domains of everyday living. Mobile health (mHealth) leverages these technologies to enhance healthcare and public health, with the potential to transform patient care, accessibility, affordability, and personalization [10]. The integration of mHealth technologies into healthcare practices is revolutionizing the way that patient care is delivered. mHealth technologies are reshaping healthcare by making medical services more accessible and efficient [11]. These tools enable patients to access care remotely, manage their health more effectively, and help healthcare providers monitor patient health in real time. This not only empowers patients but also has the potential to reduce healthcare costs and improve outcomes through better data analysis and patient management.

The acquired digital health data can be employed to develop innovative user function biomarkers, enhance the accuracy and timeliness of diagnosis, monitor treatment responses, identify individuals at risk of relapse, and provide a more objective and continuous assessment of outcomes compared to conventional methods, thereby facilitating the evaluation of intervention effectiveness [12]. The employment of mobile health applications (mHealth apps) has emerged as a promising approach to aid individuals with cognitive deficits. These apps provide interventions that leverage the capabilities of the Internet of Things (IoT) and facilitate big data analysis. By combining these technologies and focusing on vital cognitive domains, targeted mHealth apps are developed to enable the delivery of personalized and effective interventions [13]. These applications facilitate continuous evaluation and feedback, allowing users to trace their performance, identify areas for progress, and collaborate with clinicians in order to make informed decisions about their mental health.

## 2. Recent Developments in Mobile Platforms for Digital Evolution of Cognitive Assessment

Recent advances in mobile technology, particularly the integration of embedded sensors for passive data collection, have given rise to innovative mHealth applications for various health-related issues [14]. These innovations have been extensively used among older adults, addressing conditions like diabetes, chronic obstructive pulmonary disease, Alzheimer’s disease (AD), dementia, osteoarthritis, and fall risk, employing diverse sensor and implementation approaches [15,16,17,18,19]. Furthermore, the use of mobile technology, particularly in the context of mental health, highlights its versatility, support features, and adaptability in addressing various health issues, further emphasizing the potential of mHealth in healthcare. These interventions have demonstrated effectiveness in alleviating depression symptoms and addressing anxiety in adults, utilizing elements like gamification, user customization, and anonymous feedback [20,21]. Mobile-based cognitive behavioral therapy has shown promise, enhancing adherence and reducing depression symptoms, especially for individuals with limited access to traditional mental healthcare [21,22]. Additionally, mHealth leverages passive sensing technology to predict mood states and improve depression and anxiety symptoms [23], offering benefits for older adults, including those with cognitive decline. Interventions employing virtual reality, serious, and interactive video games have the potential to improve accessibility, user experience, and cost-effectiveness and provide a means of remotely administering cognitive training and rehabilitation programs while monitoring cognitive health through the utilization of new data streams and identifying indicators sensitive to detecting subtle transitions in cognitive function.

In line with the current trend of leveraging mobile and wearable digital technology, various assessments that were originally paper-based have been converted into mobile-appropriate versions, such as the eSAGE, e-CT, and eMoCA. These mobile-appropriate digital versions have demonstrated high diagnostic accuracy in distinguishing Mild Cognitive Impairment (MCI) and AD patients from healthy older adults. In addition, they offer several advantages over traditional paper-based tests, including enhanced reliability and efficiency [24,25,26,27].

However, it is essential to acknowledge the emphasis placed by researchers on the need to establish new normative standards that are specifically designed for digital assessments, particularly when validated cognitive assessments are converted into mobile-appropriate digital formats. This recommendation stems from a comparative study that revealed notable differences in both overall scores and individual item scores between mobile-appropriate digital assessments and traditional paper-based examinations [28]. These findings underscore the significance of developing appropriate benchmarks and reference points to ensure an accurate interpretation and evaluation of cognitive performance using digital assessment tools. By establishing standardized norms, the field can effectively leverage the potential of digital assessments in enhancing the assessment and management of cognitive function.

Mobile-appropriate digital versions of cognitive assessments have gained acceptance not only compared to traditional paper-based assessments but also in comparison to digital computer-administered counterparts. In particular, users report a superior experience with the digital version for mobile devices, considering their performance to be superior to the corresponding one through the evaluation conducted on desktop computers [29]. Additionally, the research emphasizes the accessibility and user-friendly nature of mobile devices, particularly for individuals facing physical challenges such as arthritis or tremors, as well as those with limited familiarity with traditional computer interfaces [29]. These findings highlight the enhanced user experience associated with mobile platforms and provide further support for the strong preference for mobile versions of cognitive assessments.

In the field of cognitive assessment, significant advancements have been made in the development of tests specifically for mobile devices and in the utilization of novel data streams [30,31,32]. These innovations aim to identify and distinguish different stages of cognitive decline [33,34,35]. Mobile screening tests and test batteries have emerged as effective tools, enabling the active participation of older adults in their cognitive assessment and monitoring [36,37,38].

Furthermore, mobile-appropriate cognitive evaluations have demonstrated strong correlation with validated cognitive assessments [34,37,38,39]. Their ability to provide high-quality cognitive screening without the need for clinician input is a notable advantage of these assessments. They also demonstrate the potential to be effectively deployed in both clinical and non-clinical settings, thereby promoting improved health outcomes and supporting individuals’ independence [39]. This feature enables their suitability for a wide range of healthcare settings, including community-based initiatives involving non-clinical personnel [39]. The integration of mobile platforms with person-centered care principles has allowed for the provision of individualized features [40]. Continuous and repeated evaluation is facilitated by suitable mobile assessments allowing for the collection of digital biomarkers of cognitive impairment, leading to the detection of patterns and changes that are often difficult to detect [41,42]. Moreover, the integration of data derived from mobile assessments with electronic health records holds promise for further enhancement of cognitive evaluation practices [37].

However, the accessibility of mHealth apps that effectively cater to a broad spectrum of cognitive domains, operate within user-selected environments and timeframes, enable unsupervised self-administration, function on a single user mobile device, and assist the purpose of training, evaluating, and monitoring, while also facilitating discreet data collection, is currently limited [13,43].

Additionally, expanding the target population beyond individuals with cognitive deficiencies to include all adults presents a significant challenge. To address this significant gap, we recently published a mHealth application, the RODI app [44], tailored to meet these multifaceted needs. The RODI app is a mobile application designed to enhance cognitive function, seeking to uplift the user’s overall well-being and psychological state (further information in the following section). In this study, we utilized the RODI app on both healthy participants and those diagnosed with NCDs to identify their unique features and the primary elements that differentiate them.

In this context, the RODI application [44] was presented, which is an intervention specifically designed for mobile devices with a dual objective. Firstly, it aims to provide process-based training to enhance function and address deficits across various domains. The application offers a user-friendly experience on mobile devices, enabling active engagement with tasks and thereby fostering motivation and enjoyment [44]. In addition, the RODI app offers several advantages in the context of assessment. It facilitates continuous and repetitive evaluations while discretely collecting data. The obtained data allow for trend identification, enabling the monitoring of changes in users’ performance. This approach provides a more comprehensive understanding of abilities and their fluctuations, which may be challenging to discern through traditional assessment methods.

Harnessing the features of the RODI app, users can enhance their monitoring and analysis of performance, resulting in more informed decision making and personalized interventions. Primarily targeting older adults with cognitive deficits, the app is also suitable for healthy adults aiming to assess and enhance their reserve. It allows individuals to self-administer cognitive tasks in their preferred environments and timeframes. Operating on a single mobile device and presenting a variety of tasks and difficulty levels, the RODI application intends to offer users a challenging and stimulating experience that supports cognitive health.

The RODI app was developed using a user-centered approach, which involved pilot testing the application in a sample of cognitively healthy adults ranging in age from 21 to 88 years old. The feedback collected from this pilot testing was then used to refine and optimize the application to better meet the needs of its users. The app’s pilot usage revealed that adults were willing to use mobile technologies for cognitive stimulation, with participants acknowledging the importance and convenience of such technologies for cognitive awareness [44].

## 3. Materials and Methods

### 3.1. Case Study Description

Having originally pioneered the RODI mHealth app as an innovative approach for NCD detection, we assess its effectiveness and explore its full potential. We employed the RODI app in an extensive, methodologically robust study that encompassed a diverse cohort of 182 participants, including both individuals diagnosed with NCDs and healthy controls (Figure 1). Our primary objective was twofold: firstly, to harness advanced machine learning techniques to analyze the data, aiming to discern clear patterns and indicators representative of NCDs; and secondly, to interpret these findings biologically, providing a holistic understanding of the underlying mechanisms. Our aim is also to exploit the insights from this study to guide the evolution of the RODI app. Our ambition is to integrate the knowledge acquired, refining the app into a more robust version enhanced by artificial intelligence. Although RODI’s cognitive tasks align with the standard criteria for NCDs [45], we believe that AI can contribute significantly to building an upgraded version for better detection of early signs of NCDs. In the subsequent subsections, we provide a detailed elaboration of the experimental study methodology and findings.

### 3.2. Participants

Several elderly care centers, open care centers for the elderly, day care centers for patients with dementia, and workplaces were approached for the recruitment of participants. The potential participants were introduced to the RODI mHealth app and the study’s intentions, and they were informed about the research approval obtained from the Ionian University’s Research Ethics and Deontology Committee (protocol number 3600). The research population included patients diagnosed with NCDs and healthy participants. This research was conducted from July 2022 to October 2022.

A total of 182 individuals participated in the study, with non-randomized and non-blinded selection criteria. Among the participants, eight (8) individuals had been diagnosed with NCD, including diagnoses of subjective cognitive decline, mild cognitive impairment, early dementia, and moderate dementia. The study included 99 female participants (54.4%) and 83 male participants (45.6%). The mean age of the participants was 46.1 years, with a mean of 15.58 years of education. The age of the participants ranged from 19 to 89 years old. Out of the total participants, 34 individuals (18.68%) were retired, while 148 individuals (81.32%) were still employed.

### 3.3. RODI App Execution

The mHealth RODI app comprises ten cognitive tasks, including three that present alternative response formats in either images or words. Additionally, one of these tasks is re-evaluated twice, increasing the total number of tasks to fifteen. The Back Task is intended to test the user’s ability to recall a displayed integer and enter it via a keypad, with recall tested one and three tasks later. The New Object Task presents the user with a set of images, with one replaced, and requires identification of the new item. In the Object Series Task, the user is requested to identify the original items in a set of objects or words. The Colours And Shapes Task is a speeded selection task of a specific combination (shape/color). The Calculating Task presents numerical values as images for simple arithmetic operations. In the Shapes Task, the user must identify the original location of a missing colored shape. The Colour-Shape Task involves the user matching a colored shape to columns of different shapes and colors. The Descending Integer Task presents the user with three consecutive numbers in a descending sequence, with the user asked to calculate the next or second term. In the Name Task, the user is presented with a name in capital letters made up of straight segments and is required to find the number of segments used. The Who Task requires the user to match images of individuals with their corresponding names, with different shots of the images shown. The user’s choices and responses, time taken, and accuracy are tracked and scored. Difficulty levels in all tasks increase with a growing number of elements or complexity, which helps to continuously challenge and stimulate users’ cognitive abilities.

We utilized a tailored lighter version of the RODI app, specifically designed for one-time use per participant on personal or shared Android^®^ mobile devices. The app was modified to a single protocol configuration to ensure consistency, operating in default Practice mode without a time limit, and displaying the correct answers after each task. Tasks were presented in random order with levels of difficulty alternating between Easy and Medium to accommodate this extended demographic of participants. To ensure variability in the required answer formats, tasks with different conditions, such as providing an image or a word as the answer, were presented in rotation. This approach allowed participants to encounter diverse formats in each task, contributing to a more comprehensive cognitive assessment and reducing potential biases in the data. The app’s features were adjusted, disabling unnecessary features, while retaining essential functions like Sign In, Practice mode, Show Correct Answer, and Show Results.

Data collection was conducted utilizing Firebase Cloud Storage. This approach enabled discrete data collection while restricting access to authenticated users only. Participants granted digital consent before creating an account and receiving comprehensive information regarding the research’s objectives, data collection procedures, anonymization processes, and subsequent analysis. During the study, several tablets were employed, each equipped with the pre-installed lighter configuration of the RODI app.

### 3.4. Methodology

We employed a Machine Learning (ML) workflow to analyze the data collected from the RODI application. ML algorithms were utilized to automatically discover patterns and relationships in the data, offering a unique perspective that can uncover non-linear relationships often missed by traditional statistical models. This ML approach is particularly advantageous for complex and unstructured data, providing valuable insights that are difficult to obtain through conventional statistical analysis. Leveraging ML techniques to analyze data collected from smartphones and tablets has the potential to significantly enhance the user experience, customizing it to individual needs and maximizing efficiency.

#### 3.4.1. Dimensionality Reduction Techniques for 2D Data Visualization

The study specifically used dimensionality reduction techniques to project the RODI application’s responses, aiming to differentiate between healthy individuals (Health state) and those diagnosed with Neurocognitive Disorders (NCD state). This visualization approach enabled the identification of patterns and trends within the data, potentially leading to the discovery of biomarkers and therapeutic targets. Three dimensionality reduction techniques, including t-SNE [46], UMAP SNE [47], and PCA SNE [48], were employed to assess the separability of the RODI outcomes, providing insights into the performance and predictive power of the reduced data.

#### 3.4.2. NCD Prediction Performance

The study aimed to evaluate the prediction performance of RODI outcomes in binary classification, distinguishing healthy individuals from those with NCDs. To achieve this, five classifiers were employed: k-Nearest Neighbors (k-NNs), Support Vector Machines (SVMs), naive Bayes, decision trees, and Random Under Sampling and Boosting (RUSBoost) [49]. The latter was selected as the majority of samples belonged to the healthy class. K-NN is a supervised learning algorithm that classifies a new observation based on its nearest neighbors. SVMs find the best boundary to separate different classes. Naive Bayes is a probabilistic classification algorithm based on the Bayes theorem. Decision trees create a tree-like model of decisions, and RUSBoost is an ensemble method combining random under-sampling and boosting techniques to handle imbalanced datasets.

#### 3.4.3. Feature Importance

In our experimental study, 182 participants engaged with the RODI mHealth app, completing 15 brief tasks, which resulted in a 182 × 15 data matrix. We implemented an ensemble feature selection strategy aimed at identifying the most dominant tasks regarding NCD identification. Our unique framework relies on the Variable Importance (VI) measure, extracted from three state-of-the-art classifiers: XGBoost, LightGBM, and CatBoost. We then systematically prioritized these 15 features by employing the Borda count. The integration of machine learning within our method unfolds immense potential, offering both precision and depth to our results. Machine learning, especially with robust classifiers like XGBoost [50], LightGBM [51], and CatBoost [52], can discern intricate patterns within large datasets, revealing relationships that might be obscured for conventional methods. In essence, the adoption of such an ML-driven approach not only amplifies the accuracy of our findings but also provides a richer, deeper understanding of the data, enabling us to draw more nuanced insights from the RODI app’s results.

The logic behind this approach is to leverage the strengths of each classifier to gain a comprehensive understanding of the data. These classifiers evaluate the tasks based on their ’Variable Importance’ (VI), which is a measure indicating how much each task contributes to the accuracy of the model in predicting NCDs. Feature selection is a critical process in machine learning that involves selecting a subset of relevant features (or variables) for use in model construction. Removing irrelevant features can lead to improved accuracy for machine learning models. Also, by eliminating redundant or irrelevant features, feature selection helps to reduce the chances of a model overfitting to the training data. More specifically, XGBoost operates on gradient-boosted decision trees. The variable importance in XGBoost is calculated by taking into account the number of times a variable appears in a tree across the ensemble of trees. Mathematically, for a given feature *f*:(1)$$VI_{XGBoost}\left(f\right)=\sum_{t=1}^{T}I\left(f\;\mathrm{appears}\;\mathrm{in}\;\mathrm{tree}\;t\right),$$
where *T* is the total number of trees and *I* is the indicator function. LightGBM, another gradient-boosting framework, computes feature importance by considering two aspects: “split” and “gain”. While “split” counts the number of times a feature is used in a model, “gain” measures the contribution brought by a feature to the model. For a feature *f*:(2)$$VI_{\mathrm{Light}\;GBM}\left(f\right)=\sum_{t=1}^{T}\;\mathrm{Gain}\;\left(f\;\mathrm{in}\;\mathrm{tree}\;t\right),$$
where Gain represents the improvement in accuracy brought by a feature. CatBoost, a boosting algorithm built on categorical features, determines feature importance by tracing back through the trees in the model and measuring how much each split improves the loss function. For feature *f*:(3)$$VI_{\mathrm{CatBoost}\;}\left(f\right)=\sum_{t=1}^{T}\Delta \;\mathrm{Loss}\;\left(f\;\mathrm{in}\;\mathrm{tree}\;t\right),$$
where Δ Loss represents the change in the loss function due to the inclusion of the feature in tree *t*. From the above three classifiers, we exported the VI score for each task within RODI, reflecting their capability to differentiate between the NCD and Health states. Then, we applied the Borda count method [53], a rank-based approach that is employed to prioritize and rank features. In our context, for each classifier, features were ranked based on their variable importance, with the most important feature receiving the highest rank (e.g., 15 for the most important, 1 for the least). These ranks were then aggregated across all classifiers. For a feature *f*:(4)$${\mathrm{Borda}}_{\mathrm{count}\;}\left(f\right)=\sum_{c=1}^{C}{\mathrm{Rank}}_{c}\left(f\right),\;$$
where ${\mathrm{Rank}}_{c}\left(f\right)$ is the rank of feature *f* in classifier *c* and *C* is the total number of classifiers. The features were then prioritized based on their cumulative Borda count, with higher counts indicating greater overall importance across classifiers.

## 4. Results

Before delving into the intricacies of machine learning, it is imperative to first understand the inherent nature and behavior of our data. Thus, we initiate a primary exploration focusing on the 15 feature tasks derived from the RODI app. By doing so, we aim to establish a foundational perspective on the inherent information within the data and discern any preliminary patterns or distinctions, setting the stage for subsequent, more advanced analyses.

Our first visualization, a violin plot (Figure 2), exhibits the distribution of each of the 15 feature tasks across the two states: Health and NCD. At a cursory glance, the plot illustrates a complex dataset where the boundaries between the two states are not distinctly demarcated. Such overlaps and intricacies in the data distribution underscore the need for a more sophisticated approach, positioning machine learning as an invaluable tool in extracting meaningful insights from such intricate datasets.

Subsequently, we turn our attention to a heatmap showcasing the correlations among the feature tasks (Figure 3). Interestingly, the heatmap does not highlight any significant correlations among the tasks. This lack of strong correlations further complicates the dataset, making traditional analysis techniques less effective in distilling knowledge. This emphasizes the challenge in our dataset, reinforcing the perspective that machine learning, with its advanced algorithms, stands as one of the few methods adept at effectively navigating and extracting meaning from such multifaceted data landscapes.

The findings (Figure 4) suggest a noteworthy degree of overlap between the Health and NCD categories, which poses a considerable challenge in the clear differentiation of individuals. This intricate data complexity underscores the multifarious nature of the problem, emphasizing the requirement for machine learning techniques capable of handling the aforementioned limitations.

Concerning the prediction performance, a 10-fold cross-validation technique was employed to evaluate the performance of the algorithms. The models were trained and tested 10 times independently to ensure the robustness of the results. The default parameter settings of each algorithm were utilized; however, a significant improvement in the performance metrics was observed through the variation in the parameters. All algorithms were examined using six measures: balanced accuracy, F1-score, specificity, sensitivity, precision, and false positive rate (Figure 5). Balanced accuracy was utilized as a metric to address class imbalance in the dataset by averaging the recall rate of each class. It is particularly useful for binary classification problems with under-represented classes. The F1-score is a measure of model accuracy that considers the harmonic mean of precision and recall. Precision is the ratio of true positive instances to the total number of true positive and false positive instances, while recall is the ratio of true positive instances to the total number of true positive and false negative instances. The F1-score is beneficial for imbalanced datasets because it equally emphasizes precision and recall. Specificity measures a classification model’s ability to correctly identify negative instances in binary classification problems. It is defined as the proportion of True Negatives (TNs) out of all instances that are negative (TN + FP). Lastly, the false positive rate indicates the proportion of negative instances that were misclassified as positive. By examining all these measures, we adopted a systems view of the evaluation, ensuring a holistic understanding of the algorithm’s performance across various facets of classification.

Our study revolves around a crucial factor, a novel ensemble method, which isolates the most critical elements in the classification of NCDs. The results of our analysis, which ranks the 15 tasks from the RODI mHealth app based on their importance in NCD identification using the Borda count ranking system and Variable Importance (VI) measures from XGBoost, LightGBM, and CatBoost classifiers, are presented in Figure 6. This ranking emphasizes the significance of tasks related to visual working memory in distinguishing between healthy individuals and those with NCDs.

## 5. Discussion

The escalating global concern around Neurocognitive Disorders (NCDs) underscores the importance of early and effective detection methods. Early identification of NCDs is not just medically pertinent but is pivotal for optimizing therapeutic outcomes. With the evolution of technology, mobile health applications (mHealth apps) have burgeoned as potential tools to cater to this need, especially for those with cognitive deficits. In this evolving landscape, our study presents the RODI mHealth app, a tool that we developed to cater to this very demand. Designed in alignment with the diagnostic criteria for NCDs, the RODI app employs a series of concise tasks as an innovative approach to discern NCD patterns. Drawing from a sample of 182 participants, both with NCDs and healthy individuals, our comprehensive study ventured to probe the app’s efficacy in differentiating between the two groups.

The focus was not only on the app’s diagnostic ability but also on streamlining the app’s features, identifying those most indicative of NCDs. To achieve this granularity, we leveraged an ensemble approach, utilizing the feature importance metrics from three cutting-edge classification algorithms. Our findings spotlighted tasks related to visual working memory as the most pivotal in distinguishing between healthy individuals and those with NCDs. Conversely, tasks related to mental calculations, executive working memory, and recall proved less essential. Interestingly, our results showed that participants, regardless of their technological proficiency and background, could navigate the app effectively, hinting at its user-friendly design.

The RODI app stands as a testament to the potential of mHealth apps in the sphere of NCD screening. Its ability to segregate healthy adults from those with NCDs amplifies its relevance as a screening tool, making it a promising candidate for future adaptations and wider clinical applications. This study sets the stage for future research, serving as a reference for mHealth app developments aiming to enhance the identification of digital markers for disorders and allied conditions. In the evaluation of performance differences between cognitively healthy adults and NCD patients, the feature importance scores obtained from the random forest algorithm were utilized to identify the critical features that differentiate between the two groups. The aim was to determine the most important characteristics that predict the outcome of interest, and the identified features were used to distinguish between Health and NCD conditions.

Furthermore, the study’s findings indicate that the top five tasks that received the highest importance scores are dependent on visual working memory abilities. These results underscore the critical role of evaluating visual working memory skills to identify cognitive impairments and aid in diagnosing and tracking cognitive decline. These findings align with earlier research conducted on both individuals who are cognitively healthy and those who have cognitive disorders. The accurate recall of fundamental visual features, such as color, shape, brightness, size, orientation, and texture, heavily relies on memory capacity and the ability to associate them correctly in memory. The developmental trajectory of visual working memory abilities shows a peak around the age of 20, followed by a gradual decline. Immediate visual memory is poorer in individuals aged 55 and above compared to young children [54]. Additionally, aging differentially impacts short-term feature memory and binding memory, with a decline in short-term color–shape binding memory primarily attributed to a reduced capacity for retaining individual features [55].

Moreover, the Shapes Task obtained the highest importance score, surpassing the immediately following task by at least 0.6 and the third task by 1.40 in order of significance scores. This specific task combines visual working memory skills and spatial representations. The observation aligns with previous studies indicating that spatial abilities decline with normal aging, albeit not uniformly, and that visuospatial deficits can manifest in very early stages of dementia. Spatial information encompasses processes and data required for determining positions and directions in one’s surroundings, including fundamental spatial abilities like object location memory. Humans utilize two primary frames of reference, namely the egocentric and allocentric, to encode and organize spatial information in memory [56,57]. Prior studies have indicated that allocentric representations operate at a slower pace compared to egocentric ones [58]. Visuospatial information and processes are essential for non-verbal cognitive functions that involve the representation and manipulation of information spatially [57]. The decline in spatial memory associated with healthy aging is influenced by coherent factors as well as attentional and executive [59]. In the early stages of AD, visuospatial deficits manifest as impairments in various cognitive abilities, including constructive skills, visuospatial intelligence, spatial short-term memory, and spatial orientation [59]. These spatial orientation difficulties are evaluated as early signs of dementia and are frequently attributed to hippocampal damage, which is crucial for both general and spatial memory [60,61,62]. Furthermore, research findings suggest that neurodegenerative processes affect visual neural pathways, potentially contributing to declines in other cognitive domains [63]. Some researchers proposed that visuospatial deficiencies may occur very early in the course of dementia, while the allocentric component of spatial memory has been suggested as a potential predictor of AD from MCI and early-onset dementia [64,65,66]. Overall, deficits in spatial memory are crucial in understanding cognitive decline and may serve as an indicator demonstrating departure from normal aging [59].

The task ranked second in terms of importance scores demonstrated superior performance compared to the third-ranked task by a margin of 0.60. Similarly, the third-ranked task outperformed the subsequent task by approximately 0.40. In addition to evaluating visuospatial working memory skills, these tasks also assess executive function, which Miyake and Friedman [67] describe as higher-order cognitive processes that allow individuals to adapt their behavior in response to contextual changes, with a particular focus on inhibition and cognitive flexibility. Diamond’s model [68] identifies three main components of executive function, namely inhibition, working memory (including visuospatial working memory), and cognitive flexibility, that facilitate the development of complex cognitive processes. Previous research has reported deficits in inhibitory control [69,70] and partially in cognitive flexibility [71,72] in patients with AD. Additionally, it is indicated that individuals with MCI exhibit deficits in tasks that assess executive function, including inhibition and cognitive flexibility [73]. The findings suggested that these tasks may offer clinically relevant information about the decline in executive functions in individuals with MCI, providing diagnostic potential by assessing their discriminatory power. Similar results were observed in individuals diagnosed with dementia, indicating a clear executive deficit in inhibitory control and a partial deficit in cognitive flexibility [74]. The researchers suggested that there exists an executive functioning profile in AD, characterized by impairment in inhibitory control and cognitive flexibility, where performance variations may reflect differences in executive function deterioration levels during Alzheimer’s disease progression. Furthermore, the researchers noted that the use of computerized versions of these tasks provides more precise measurements of reaction times and accuracy, which are more discriminative and sensitive in detecting executive function decline, indicating their usefulness in neuropsychological batteries for MCI diagnosis.

According to the importance scores, projects that assess visuospatial abilities rank higher than other visual projects, which is consistent with the findings of Alescio-Lautier et al. [64]. In their study, they examined whether visual or visuospatial modalities were more affected in early memory impairment in Alzheimer’s disease by comparing AD and MCI patients with healthy controls. The results indicated that both MCI and AD patients exhibited impairments in short-term memory and visuospatial short-term memory when compared to the healthy control group. Notably, the impairment in spatial performance was more pronounced than in visual performance. MCI patients exhibited intermediate performance levels between the healthy controls and AD patients. Additionally, the cognitive memory profiles differed based on the modality tested, indicating distinct underlying processes. Specifically, AD patients exhibited more significant visuospatial deficits and were more impacted by experimental modifications, possibly due to a phenomenon known as the attentional blink. This phenomenon leads to temporary functional blindness when sequentially presented stimuli are presented rapidly. The researchers suggested that differences in visual recognition may result from deficits in attentional and executive resources, while scarcities in spatial recognition may be indicative of a genuine spatial disorder.

In terms of importance scores, the projects related to executive working memory rank from 8th to 12th, exhibiting a range of approximately 0.25. However, the Object Series Task (image) stands out as an exception, with an importance score exceeding 1 (namely 1.44) and securing the fourth rank. The unique characteristic in question can be attributed to the task’s demand for identifying, in a random sequence, the precise arrangement of images that do not have explicit correspondences with identifiable objects that can be linked to language. The images employed in this task derive from mahjong and domino tiles as well as playing cards and dices, thus heightening its level of difficulty. Image memory tasks are employed to evaluate executive working memory, and their potential utility in assessing individuals with MCI and AD has been extensively explored. Studies have demonstrated that AD patients perform poorly on conventional tests of executive function [75] and working memory [76], which is attributable to impairments in the central executive function [77]. The prefrontal cortex primarily processes executive control [78] and the significant accumulation of plaques in the frontal lobes observed could account for the lower executive function scores in AD patients [79,80]. According to an exploratory study that investigated EEG changes during memory tasks, including word memory, it was observed that the picture memory task showed notable EEG differences between MCI patients and controls [81]. Moreover, in a study focused on identifying a potential biomarker for distinguishing between MCI and AD, multi-domain cognitive testing was used to evaluate executive working memory in patients with MCI and AD compared to control participants [82]. The study was consistent with previous findings that executive working memory is significantly impaired in both MCI and AD patients compared to controls. Furthermore, a positive correlation was found between executive working memory tasks (word and picture memory) and MCI, suggesting that better performance in these tasks may lead to improved global cognitive scores. Executive working memory was found to be more severely affected in AD patients compared to both MCI and controls. Additionally, a negative relationship was observed between word and picture memory tasks and Clinical Dementia Rating. Among the administered cognitive tests, picture memory was found to be more robust, exhibiting high sensitivity and specificity compared to controls. These findings indicate that the picture memory task may be a useful tool for distinguishing differences among all groups and therefore may aid in the early detection of cognitive impairment, enabling timely interventions.

In the context of image selection, the Who Task holds the fifth rank in order of score importance, whereas its counterpart with name selection ranks third from the end, with a minimum difference of 0.5 between them. The formation of face–name associations is a widely recognized challenging task, particularly among older adults, who frequently report difficulties in remembering proper names [83]. Conversely, the association of a face with other biographical information, such as occupation or hobbies, is relatively easier, primarily due to the inherent unrelatedness of a face with a name [84]. This specific task lacks contextual cues, making it difficult to form an associative link between a proper name and a unique face, thereby requiring higher cognitive effort. Successful performance on face–name association tasks has been linked to increased brain activity within memory-related networks in both young and older populations [85]. The rank of the Who Task in the case of image selection is consistent with the findings of previous studies. A review of studies exploring the use of a demanding test of face–name associative memory as a tool for early diagnosis of AD concluded that it may be a valuable diagnostic tool, and its performance is related to Aβ in brain regions associated with memory systems [86]. Additionally, a computerized, self-administered test was found to be suitable for discriminating between cognitively healthy and amnestic MCI individuals [87]. Performance on this test was associated with AD cerebrospinal fluid biomarkers, enabling the detection of memory-impaired cases resulting from other aetiologies. Furthermore, the researchers stated that the test can detect the AD endophenotype and is associated with AD-related changes in MRI and cerebrospinal fluid in patients with early-onset MCI.

In terms of tasks requiring the user to input answers using the device’s built-in keyboard, the Back Task (3rd) ranked highest, appearing as the sixth task in descending order. This task involves delayed recall, occurring three tasks after the number to be recalled is presented. On the other hand, Immediate recall, Back Task (1st), ranked last, with a significance score of only 0.01. Back Task (2nd), a post-immediate recall task, was ranked eleventh in terms of importance score. These findings are consistent with prior research that has demonstrated an association between lower scores on delayed recall tasks and older age [88]. Delayed recall tasks have been shown to be the most effective neuropsychological predictors of conversion from MCI to AD [89,90]. Moreover, a study by Sano et al. [91] aimed at determining the usefulness of delayed recall assessment in clinical trials for MCI and AD showed that the addition of delayed recall increased the sensitivity to detecting changes in subjects with MCI, while it increased the variance in subjects with AD, even in those with mild impairment.

In the context of mental calculations, which involve arithmetic operations performed without the aid of devices or tools, the Calculating Task ranks seventh in terms of importance with a score of 0.73, followed by the Name Task in tenth place with a score of 0.65. Conversely, the Descending Integer Task is ranked second, but only has an importance score of 0.1. Mental calculation is a fundamental mathematical skill that is closely related to procedural fluency, one of the five main components of mathematical proficiency, which refers to the ability to perform procedures efficiently, accurately, and flexibly. According to an exploratory study that investigated EEG changes during memory tasks, including word memory, it was observed that the picture memory task showed notable EEG differences between MCI patients and controls [92,93]. With an aging population, the likelihood of cognitive deterioration is expected to increase, potentially affecting financial capabilities [94]. Such deficits in financial abilities within this demographic could also present societal risks. In particular, written arithmetical skills have been identified as the key predictor of financial capability across the various stages of dementia [95]. Patients with MCI may experience difficulties with mathematics that affect their everyday functioning. In a study investigating the brain changes and cognitive factors associated with these deficits, researchers noted that among MCI patients, issues with number comprehension and formal numerical performance were linked to variations in brain volume in the right middle occipital and right frontal gyrus region deficits [92]. These findings indicate that early neuropathological changes in various brain areas, including the frontal, temporal, and occipital regions, can lead to cognitive deterioration in MCI, affecting daily numerical functioning. As a result, they have significant implications for the diagnosis, clinical care, and at-home support of MCI patients.

Furthermore, a study investigated the numerical-information-processing ability of individuals with MCI but without dementia and those with mild dementia of the Alzheimer’s type [96]. In particular the study evaluated the capacity of patients to perform simple numerical operations without added attentional or executive load and when required to switch between functions or control and inhibit automatic retrieval processes. The results showed that both patient groups could retrieve numerical knowledge from long-term memory without added load. However, under executive load, patients with dementia of the Alzheimer’s type demonstrated compromised executive function, while patients with MCI exhibited difficulty inhibiting previously learned associations. The researchers emphasized the importance of assessing numeracy processing in a mixed condition that mimics everyday numeracy activities, as it highlights the contribution of attention and executive functions to numeracy. They also suggested that patients who score within the normal range on routine neuropsychological numerical assessments may still experience difficulties when additional non-numerical resources are required.

In summary, the study’s findings support the potential of the RODI app as a screening tool for NCD, as it was able to distinguish between healthy cognitive adults and NCD patients. The analysis also revealed the feature importance scores obtained from the random forest algorithm, indicating the most salient features that differentiate between the two situations. The “Shapes Task” feature exhibited the highest level of importance, while the “Back Task (1st)” feature displayed the lowest level of importance. The low importance of “Back Task (1st)” suggests that its exclusion would not significantly impact the overall performance of the model. The importance of certain features, specifically those related to visuospatial abilities, in discriminating between the two situations is noteworthy. The observation that the four features with importance scores above 1 are related to visual working memory further reinforces this finding. These results suggest that future research may benefit from focusing on these features.

Furthermore, future research should address the primary limitations of this study, which include various aspects. Firstly, the sample selection process was non-randomized and non-blinded, relying on a convenience sample. Additionally, the study faced challenges with a relatively small sample size, especially concerning the NCD cohort. While the preliminary findings from the current study seem favorable, there is a compelling need for further research to comprehensively establish the application’s effectiveness. The app’s tasks align with criteria for neurocognitive disorders, target related domains, and offer diverse tasks. Despite positioning RODI as a promising tool for identifying deficits, rigorous scientific investigations are necessary to confirm its validity and effectiveness.

It is worth mentioning that the naive Bayes classifier outperformed other machine learning algorithms, including RUSBoost, kNN, SVM, LDA, and decision trees, for the specific problem that we addressed. One possible explanation for the superior performance of the naive Bayes classifier is its independence assumption. This classifier assumes that features are conditionally independent given the class label. This simplification can make the naive Bayes classifier more robust and efficient when handling high-dimensional or noisy data. If our dataset exhibits relatively independent features, this characteristic may have contributed to the strong performance of the naive Bayes classifier compared to other algorithms. Additionally, the probabilistic nature of the naive Bayes classifier may be more suitable for our specific problem. As a probabilistic classifier, it can manage uncertainty effectively and provide probabilities associated with each class. These probabilities can be valuable for decision making in our application. The study’s findings provide substantial evidence supporting RODI’s potential as an assessment tool for Neurocognitive Disorders (NCDs), given its effectiveness in distinguishing between healthy adults and NCD patients. The analysis also yields importance scores, shedding light on the most critical attributes contributing to this differentiation. Notably, tasks associated with visuospatial abilities stand out as the most crucial, with the four related to visual working memory achieving the highest importance scores. These findings suggest that future research endeavors may benefit from prioritizing investigations into these specific characteristics.

There are additional areas of investigation to explore, including the use of the full version of the RODI mHealth app with its active features. Furthermore, analyzing the comprehensive dataset collected by the app, including user demographics, task completion times, screen time, and any corrections made, would provide valuable insights. Exploring the interaction of the app with other technological tools that provide biomarkers is also promising. The app’s potential as both a cognitive training intervention and an evaluation tool highlights its value for future research and clinical applications. These investigations can enhance our understanding of the app’s efficacy and its clinical utility.

RODI has the potential to be directed toward younger older adults with normal cognitive function who have not displayed any cognitive impairment. This approach can facilitate the detection of cognitive disorders in the earliest stages or can be utilized to evaluate cognitive reserve. Furthermore, RODI can be aimed at individuals who do not have access to appropriate healthcare services or those living in territories with prolonged waiting lists for such services, thus discouraging them from seeking medical care.

The engaging nature of mobile cognitive interventions and their potential of self-administered screening without imposing extra limitations on the healthcare system present an opportunity for extensive screening and alleviating the strain on primary healthcare services. Incorporating screening into apps that the elderly are inclined to use, particularly self-administered tasks, could prove to be an effective approach. However, the inclusion of such mHealth apps into healthcare systems, information policies, and awareness campaigns requires additional investigation [97]. It is also essential to study the perspectives and requirements of application users in order to develop more effective apps that encourage older adults to self-monitor and enhance their cognitive functions.

Remote assessment and cognitive training utilizing mHealth apps allow for unsupervised sessions. However, sessions conducted in the presence of clinicians provide the benefit of professional judgment. Therefore, it is essential to emphasize that these interventions are not meant to substitute for regular visits to specialists but rather to supplement them by offering data and insights into individuals’ progress and cognitive status.

## 6. Conclusions

Our study underscores the pivotal role of mHealth applications, particularly the RODI app, in the early detection and assessment of Neurocognitive Disorders (NCDs). The study’s outcomes demonstrate RODI’s effectiveness in distinguishing between NCD-affected individuals and healthy controls, emphasizing the relevance of tasks associated with visual working memory. These tasks emerged as the most critical indicators in identifying the presence of NCDs, overshadowing other cognitive tasks in predictive significance. The importance scores derived from our machine learning analysis provide a valuable roadmap for prioritizing features in future mHealth tools. Our research opens the road for the advancement of digital health technologies and sets a strategic direction for subsequent explorations into neurocognitive assessment.

## Figures and Tables

**Figure 1 healthcare-11-02985-f001:**
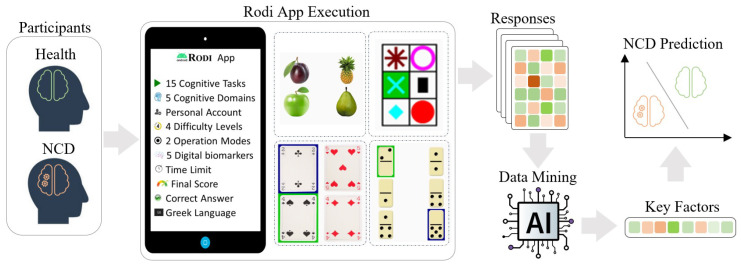
The figure illustrates the deployment of the RODI app in a robust study with 182 participants, including both NCD-affected individuals and healthy controls. Within the app, users engaged with 15 distinct cognitive tasks. We adopted an ensemble feature selection method to pinpoint the most significant tasks based on participants’ performance data. This method utilized an ensemble Variable Importance (VI) metric derived from analyses conducted by three advanced classifiers: XGBoost, LightGBM, and CatBoost. Following data collection, we ranked these cognitive tasks using the Borda count technique to determine their relative importance. The final analysis revealed the key tasks that effectively differentiate between NCD and healthy cognitive states.

**Figure 2 healthcare-11-02985-f002:**
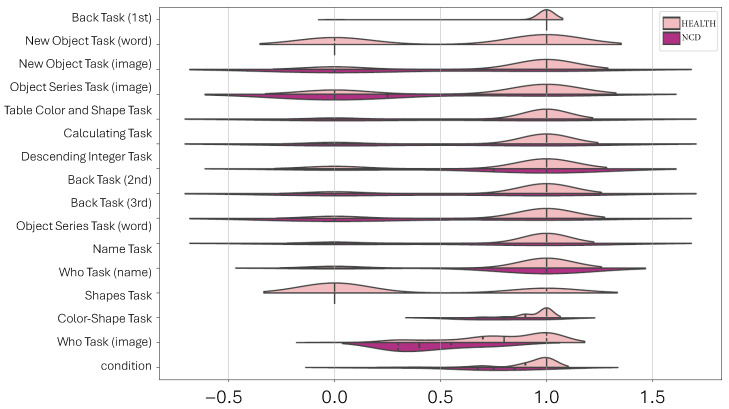
Violin plot illustrating the distribution of 15 feature tasks across two states: Health and NCD. The overlapping distributions highlight the complexity of the dataset and the challenges in distinguishing between the two states.

**Figure 3 healthcare-11-02985-f003:**
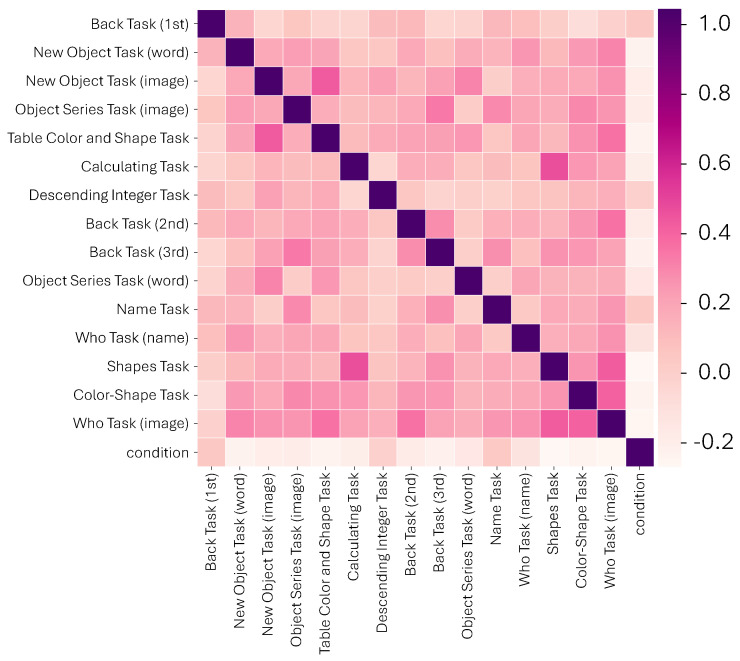
Heatmap displaying the correlations among feature tasks. The absence of significant correlations underscores the complexity of the dataset and the necessity for advanced machine learning techniques to decipher it.

**Figure 4 healthcare-11-02985-f004:**
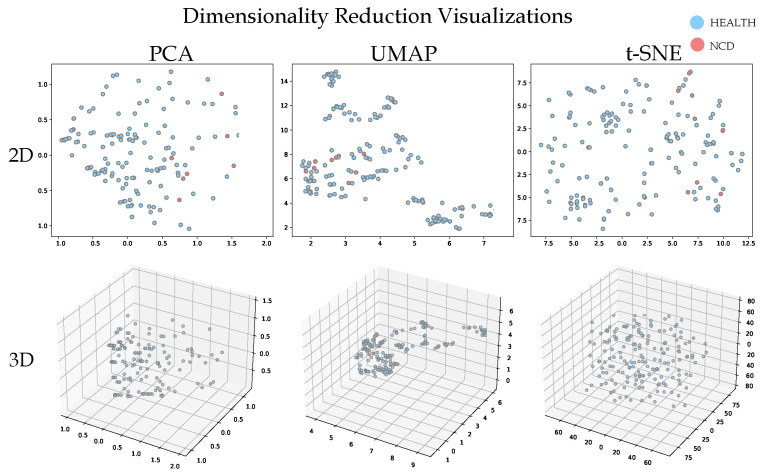
Visualization of RODI application responses using dimensionality reduction techniques: t-SNE, UMAP SNE, and PCA SNE. The projection aims to differentiate between the Health state and NCD state, highlighting patterns and potential biomarkers within the data.

**Figure 5 healthcare-11-02985-f005:**
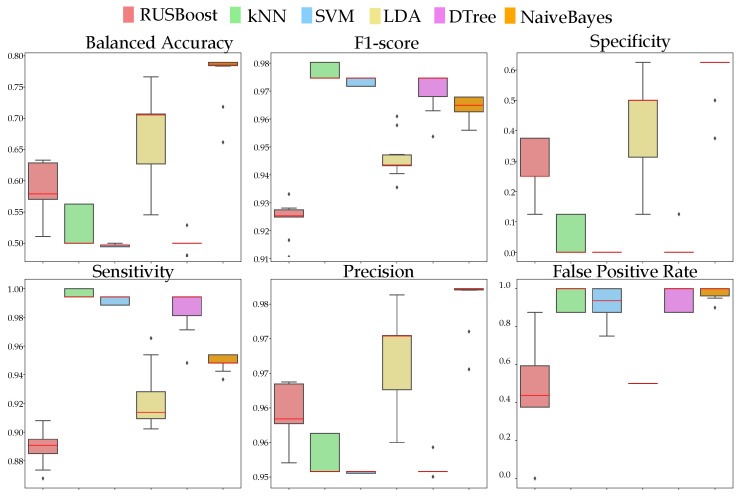
Boxplots illustrating the performance of five classifiers (k-NN, SVMs, naive Bayes, decision tree, and RUSBoost) in predicting RODI outcomes for binary classification of Health and NCD states. Using a 10-fold cross-validation, the models were trained and tested through 10 independent iterations. The evaluation assessed balanced accuracy, F1-score, specificity, sensitivity, precision, and false positive rate.

**Figure 6 healthcare-11-02985-f006:**
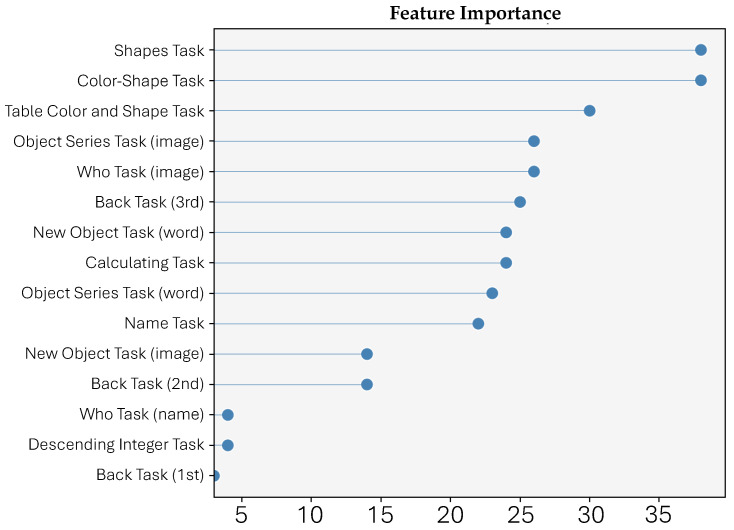
Dot plot ranking of the 15 tasks from the RODI mHealth app based on their importance in NCD identification. Features were prioritized using the Borda count ranking system, derived from the Variable Importance (VI) measure of three classifiers: XGBoost, LightGBM, and CatBoost. Our analysis revealed that tasks related to visual working memory were the most significant in distinguishing between healthy individuals and those with NCD. In contrast, processes involving mental calculations, executive working memory, and recall were less influential. This ensemble feature selection strategy showcases the potential of machine learning in discerning key patterns for NCD detection.

## Data Availability

The data for the experiment can be accessed online at: https://www.dropbox.com/scl/fo/vx7ovu6i8uaqi9maaqqpm/h?rlkey=5gkeyaularf0vteszcna5sggw&dl=0.

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
