# Peer review of "The RODI mHealth app Insight: Machine-Learning-Driven Identification of Digital Indicators for Neurodegenerative Disorder Detection"

_healthcare, 2023, doi:10.3390/healthcare11222985_

Round 1

Reviewer 1 Report

Comments and Suggestions for Authors

This paper present trending topic of  using ML in future mHealth to detect cognitive disorder  but the work consists of a technical content rather than scientific content.  Linking between manuscript sections are lacking and need improvement. Thank you 

Author Response

Dear Reviewer,

We appreciate your comments regarding the emphasis on technical content in our manuscript. We acknowledge that the scientific underpinnings of our work may not have been as prominently highlighted as they should have been. In the computational section of our paper, we introduce a novel hybrid ensemble learning approach for feature selection. This method is specifically tailored to discern the most significant and impactful features within our dataset, which is crucial for the detection of cognitive disorders in the mHealth domain. In the mental health realm, we have designed and implemented a distinctive approach that addresses three critical aspects of Neurocognitive Disorders: assessment, monitoring, and training. This tripartite strategy is part of what makes our contribution unique to the field.

In the revised version of our manuscript, we have made a concerted effort to elucidate the motivation and originality of our work. We believe these enhancements will more clearly demonstrate the strong scientific content and contribution of our research.

We are grateful for the opportunity to refine our manuscript.

Reviewer 2 Report

Comments and Suggestions for Authors

An interesting research article, part of an approach designed to enrich scientific knowledge.

The authors could reintegrate some sources, such as

Zhang, M. W., & Ho, R. C. (2017). M-health and smartphone technologies and their impact on patient care and empowerment. The Digitization of Healthcare: New Challenges and Opportunities, 277-291.

Marsch, L.A. (2021). Digital health data-driven approaches to understand human behavior. Neuropsychopharmacology46(1), 191-196.

Some parts of the articlespecially the introduction, should incorporate elements focusing on the adoption and usage in M-health, before delving into the specific subject of the study.

The methodology requires a comprehensive revision. It currently does not highlight the value of the research and should be presented with more precision. The presentation of the results and the discussion also need improvement. These three aspects must be revisited to enhance the quality of the proposed article.

Comments on the Quality of English Language

An interesting research article, part of an approach designed to enrich scientific knowledge.

The authors could reintegrate some sources, such as

Zhang, M. W., & Ho, R. C. (2017). M-health and smartphone technologies and their impact on patient care and empowerment. The Digitization of Healthcare: New Challenges and Opportunities, 277-291.

Marsch, L.A. (2021). Digital health data-driven approaches to understand human behavior. Neuropsychopharmacology46(1), 191-196.

Some parts of the articlespecially the introduction, should incorporate elements focusing on the adoption and usage in M-health, before delving into the specific subject of the study.

The methodology requires a comprehensive revision. It currently does not highlight the value of the research and should be presented with more precision. The presentation of the results and the discussion also need improvement. These three aspects must be revisited to enhance the quality of the proposed article.

Author Response

We greatly appreciate the reviewer's constructive feedback and suggestions. The comments provided have significantly enriched our manuscript, and we express our sincere gratitude for such valuable insights. We have addressed each of the reviewer's comments and have outlined our responses to them individually below.

Comment #1

The authors could reintegrate some sources, such as

  • Zhang, M. W., & Ho, R. C. (2017). M-health and smartphone technologies and their impact on patient care and empowerment. The Digitization of Healthcare: New Challenges and Opportunities, 277-291.
  • Marsch, L.A. (2021). Digital health data-driven approaches to understand human behavior. Neuropsychopharmacology46(1), 191-196.

Reply:

  1. I) We have carefully integrated the recommended sources. These references have been added to the revised manuscript to enrich the context and support our findings.

Comment #2

Some parts of the articlespecially the introduction, should incorporate elements focusing on the adoption and usage in M-health, before delving into the specific subject of the study.

Reply:

We are grateful for the insightful comment highlighting the need to focus on the adoption and usage of mHealth in the introductory section of our article. The manuscript has been updated to include a thorough discussion of how mHealth technologies are being integrated into healthcare practices and their impact on patient care. This addition ensures that readers are well informed about the general context of mHealth before we transition into the specific subject of our study.

Comment #3

The methodology requires a comprehensive revision. It currently does not highlight the value of the research and should be presented with more precision. The presentation of the results and the discussion also need improvement. These three aspects must be revisited to enhance the quality of the proposed article.

Reply:

We concur with the feedback regarding the methodology, presentation of results, and discussion in our manuscript. In response, we have undertaken a thorough revision to address these concerns. We have enhanced the clarity and precision of our methodology to better emphasize the value of our research. Additionally, we have refined the structure of our results and discussion sections to provide a more coherent and impactful narrative. These modifications have been made with careful consideration of the highlighted points, ensuring that our revised manuscript now aligns with the recommendations of this valuable comment.

Reviewer 3 Report

Comments and Suggestions for Authors

This study tested a smartphone app for detecting NCD people, which is very meaningful for global mental health. I have several comments,

1.      Please explain term RODI when it first appeared in the paper.

2.      I am concerned whether this app can detect specific types of NCDs or just for general cognitive impairments.

3.      The review of the global trials of such apps is not enough. As far as I know, research teams in many countries have developed apps to detect mental health issues such as anxiety and depression.

4.      Please rearrange the structure of the third chapter which should be “Methods”.

5.      Figure 1 needs more explanations.

6.      I am still confused about the detecting process. Please use a flow chart to show the whole process of identifying NCD features.

7.      How did you avoid the sample bias of your participants ranged from 19 to 89.

Author Response

Comment #1

  1. Please explain term RODI when it first appeared in the paper.

Reply:

Apologies for any confusion regarding the term "RODI." It appears we may have inadvertently given the impression that it is an acronym. We did not clearly explain its origin previously. "RODI" is not an acronym. More specifically, RODI is the name we have chosen for the app for its symbolic significance, deeply entrenched in Greek tradition. The pomegranate, or "ρÏŒδι" in Greek, stands as an emblem of fertility, abundance, rebirth, and immortality—concepts that align closely with the app's goals of cognitive renewal and enhancement.

We have provided the necessary citations for our app called RODI, designed with the intention to enrich our understanding and management of cognitive health.

Comment #2

  1. I am concerned whether this app can detect specific types of NCDs or just for general cognitive impairments.

Reply:

We appreciate your valuable feedback and would like to point out that our manuscript includes a section that addresses your concern. Specifically, the capability of the app to identify particular types of Neurocognitive Disorders (NCDs), as opposed to its use for general cognitive impairments, is an issue that we recognize due to several fundamental limitations discussed in our study.

We must, however, emphasize the preliminary nature of our findings given the primary limitations highlighted within our study. These include the non-randomized, non-blinded selection of a convenience sample and the relatively small size of our NCD cohort. While initial results are promising, they underscore the necessity for more extensive research to validate the app’s efficacy in detecting specific NCDs conclusively. The app includes tasks that are theoretically grounded in the criteria for neurocognitive disorders. Although this positions RODI as a potentially valuable tool for cognitive deficit screening, the validation of its precision requires rigorous scientific scrutiny. We aim to expand our sample in future studies to encompass a broader spectrum of cognitive impairments. This will be critical to fine-tuning the app’s utility for different user groups.

Furthermore, longitudinal studies are paramount for assessing the app’s capability in tracking cognitive changes over time. We are committed to pursuing this research direction, as it will be instrumental in establishing the app’s role in the precise identification of NCDs. Only through continued research and iterative improvements, guided by our study's outcomes, can we fully respond to the question of the app’s specificity in detecting NCDs.

Comment #3

  1. The review of the global trials of such apps is not enough. As far as I know, research teams in many countries have developed apps to detect mental health issues such as anxiety and depression.

Reply:

We sincerely appreciate your valuable input regarding the scope of our review pertaining to global trials of comparable applications. In response to your feedback, we have revised the manuscript to encompass an in-depth discussion of the integration of mHealth technologies, specifically within the realm of mental health, including the detection and management of conditions such as anxiety and depression, and their transformative implications for patient care. Your comment has significantly enriched the quality of our work, and we thank you for your constructive engagement with our research.

Comment #4

  1. Please rearrange the structure of the third chapter which should be “Methods”.

Reply:

Thank you for the suggestion to restructure the third chapter. We have revised the chapter’s organization to better align with the reviewer's preferences. However, we have chosen not to label this section simply as “Methods” to avoid a monotonous tone due to the extensive and varied pipeline our research encompasses. We believe that the new structure meets the reviewer's expectations and facilitates a clearer understanding of our research process.

Comment #5

  1. Figure 1 needs more explanations.

Reply:

We have now included a detailed caption that provides a thorough explanation of the processes and results depicted in the figure.

Comment #6

  1. I am still confused about the detecting process. Please use a flow chart to show the whole process of identifying NCD features.

Reply:

We apologize for any confusion that may have occurred. Please refer to Figure 1, which includes a flow chart representing the entire process of identifying NCD features. With the updated and detailed caption for the figure, we believe its contents will now be comprehensible. The key aspect to understand is that we collected data from both healthy individuals and those with NCDs using the 15 cognitive tasks available in the RODI app. Through our unique machine learning technique in the domain of feature selection, we were able to rank the importance of each feature. This allowed us to determine the most critical factors among the various cognitive tasks.

Comment #7

  1. How did you avoid the sample bias of your participants ranged from 19 to 89.

Reply:

Thank you for highlighting the importance of considering age range in our study. We concur that the inclusion of a broad age spectrum can indeed enhance the generalizability of our findings, providing valuable insights into cognitive performance across the adult lifespan. This methodological choice contributes significantly to the robustness of our data and the applicability of the RODI app to a diverse user base.

We acknowledge the potential impact of having a larger patient cohort to more finely differentiate the effects of age. In an ideal scenario with ample participants, a more nuanced age stratification would allow for an even more precise analysis of age-related cognitive patterns.

It is also important to note that our participant pool did include a smaller proportion of younger individuals. We made a conscious decision not to exclude these data points out of concern that it would reduce the overall dataset size. This decision was not taken lightly; we carefully considered the trade-off between dataset size and potential bias. We believe that retaining these participants, despite their smaller numbers, adds value to our preliminary findings without significantly skewing the results.

Moving forward, we aim to expand our participant base, which will naturally allow for a more detailed age stratification and the ability to control more rigorously for age-related factors. We appreciate your insight on this matter and will certainly consider it in the design of future studies.

Reviewer 4 Report

Comments and Suggestions for Authors

Thank you for the opportunity to review the manuscript. The authors studied, described and discussed pertinent information about the use of the RODI application in the detection of neurodegenerative diseases. Please find my suggestions.

ABSTRACT

Authors could include the date and objectives of the study.

STRUCTURE OF THE TEXT

The structure of the text is confusing. However, almost all the necessary information is described.

-In fact, “2. Digital Evolution in Cognitive Assessment – Recent Mobile Platforms” could be an item in the Introduction.

-The objectives should be described in the Introduction (and just once in the text): “The study aimed to (a) evaluate performance differences between healthy older adults and patients diagnosed with NCD, (b) identify significant performance differences between the two groups during the initial administration of the RODI app, and (c) determine the most critical features for predicting the outcome of interest.” (Lines 185-188).  However, we can read in different parts of the text:

(Lines 61-64): “Through traditional machine learning methods, we aim to uncover insights that conventional statistical and data analyses might miss. This research aims to contribute to the current knowledge in the field and lay the foundation for future advancements in timely assessing cognitive decline.”

(Lines 155-156): “To validate the efficacy of the RODI app and to delve deeper into its potential, we began an in-depth study.”

(Lines 157-162): “Our primary objective was twofold: firstly, to harness advanced machine learning techniques to analyze the data, aiming to discern clear patterns and indicators representative of NCDs; and secondly, to interpret these findings biologically, providing a holistic understanding of the underlying mechanisms. Our aim is also to exploit the insights from this study guiding the evolution of the RODI app.”

(Lines 185-188): “The study aimed to (a) evaluate performance differences between healthy older adults and patients diagnosed with NCD, (b) identify significant performance differences between the two groups during the initial administration of the RODI app, and (c) determine the most critical features for predicting the outcome of interest.”  Why does this paragraph come after the results?

-The authors described the methodology and results together. The authors could describe the “Materials and Methods” section. In another section ("Results" section), they could write what they found in their study.

- The "Conclusion" section is too long. Lines 599-638 could be in the "Discussion" section.

ADDITIONAL INFORMATION TO BE INCLUDED

-(Lines 83-85): “These mobile appropriate digital versions have demonstrated high diagnostic accuracy in distinguishing MCI and AD patients from healthy older adults.” The authors could describe the acronyms "MCI" and "AD".

-(Lines 174-176): “The research population included adults with normal cognitive status for their age, adults experiencing cognitive deficits without a diagnosis, and patients with diagnosed cognitive impairments.”  The authors could describe how "adults experiencing cognitive deficits without a diagnosis” were thus disclosed.

Author Response

Comment #1

ABSTRACT

Authors could include the date and objectives of the study.

Reply:

We have incorporated the date and objectives of the study into the manuscript for clarity and specificity.

Comment #2

STRUCTURE OF THE TEXT

The structure of the text is confusing. However, almost all the necessary information is described.

Reply:

We apologize for any confusion caused by the structure of our text. We have made an effort to address each comment individually and have restructured the manuscript to ensure clarity and coherence in presenting all the necessary information.

Comment #3

-In fact, “2. Digital Evolution in Cognitive Assessment – Recent Mobile Platforms” could be an item in the Introduction.

Reply:

We have reevaluated and modified the structure of our manuscript. We believe these changes have made the structure clearer and less confusing for the reader.

Comment #4

-The objectives should be described in the Introduction (and just once in the text): “The study aimed to (a) evaluate performance differences between healthy older adults and patients diagnosed with NCD, (b) identify significant performance differences between the two groups during the initial administration of the RODI app, and (c) determine the most critical features for predicting the outcome of interest.” (Lines 185-188).  However, we can read in different parts of the text:

Reply:

We apologize for the repetition of the study's objectives in various parts of the text. We have updated the manuscript to ensure that the objectives are described succinctly and solely. Thank you for bringing this to our attention.

Comment #5

-The authors described the methodology and results together. The authors could describe the “Materials and Methods” section. In another section ("Results" section), they could write what they found in their study.

Reply:

We have separated the "Materials and Methods" section from the presentation of results, as you suggested. The "Results " section now contains the findings of our study, contributing to a more systematic presentation of the research process.

Comment #6

- The "Conclusion" section is too long. Lines 599-638 could be in the "Discussion" section.

Reply:

We have revised the "Conclusion" section to be more concise, transferring the detailed content from lines 599-638 to the "Discussion" section for better coherence and flow of the manuscript. Thank you for the suggestion.

ADDITIONAL INFORMATION TO BE INCLUDED

Comment #7

-(Lines 83-85): “These mobile appropriate digital versions have demonstrated high diagnostic accuracy in distinguishing MCI and AD patients from healthy older adults.” The authors could describe the acronyms "MCI" and "AD".

Reply:

Thank you for bringing this to our attention. We have revised the manuscript, and now, the acronyms MCI and AD are fully spelled out upon their first appearance. However, it is worth noting that the term "AD" appears for the first time earlier in the updated version of the text, separate from its second mention alongside "MCI."

Comment #8

-(Lines 174-176): “The research population included adults with normal cognitive status for their age, adults experiencing cognitive deficits without a diagnosis, and patients with diagnosed cognitive impairments.”  The authors could describe how "adults experiencing cognitive deficits without a diagnosis” were thus disclosed.

Reply:

Apologies for any confusion caused; there was a misplacement of that term in our document. What we intended to convey is that our analysis suggested some health profiles were similar to those found in NCDs. Therefore, one hypothesis could be that these adults might be approaching an NCD state. However, since this remains a hypothesis and not a confirmed participant characteristic, we have opted to exclude it from the description of our participant group. We appreciate your attention to this detail and thank you for bringing it to our attention.

Round 2

Reviewer 1 Report

Comments and Suggestions for Authors

Revised version is much better, but methodology still complicated for casual readers or healthcare practitioners with less experience in ML. 

Comments on the Quality of English Language

Few grammar mistakes need to be correct as well abbreviations all over the manuscript. 

Author Response

Thank you for your valuable feedback on the 2nd revised version of our manuscript. We appreciate your observation regarding the complexity of the methodology section, especially for casual readers or healthcare practitioners less experienced in machine learning. To address this, we further simplified and clarified the methodology, especially the Feature Selection strategy. We believe these improvements will make our manuscript more inclusive and understandable to a broader audience.

Also, we have thoroughly reviewed the document and made the necessary corrections to address these issues. We have rectified the grammatical errors to ensure the text is clear and concise.
